# Influence of Hydrogen Reduction on the Properties of Porous High-Nitrogen Austenitic Stainless Steel

**DOI:** 10.3390/ma15165585

**Published:** 2022-08-15

**Authors:** Weipeng Zhang, Liejun Li, Tungwai Ngai, Ling Hu

**Affiliations:** Guangdong Key Laboratory for Processing and Forming of Advanced Metallic Materials, South China University of Technology, Guangzhou 510640, China

**Keywords:** powder metallurgy, nitrides, hydrogen reduction, corrosion resistance

## Abstract

This work explores the impact of hydrogen reduction on sintering and nitriding of porous high-nitrogen austenitic stainless steel (HNASS) processed via powder metallurgy. A temperature-resolved hydrogen reduction (temperature range of 700–1250 °C) was performed to evaluate the phase composition of porous HNASS. The systematic microstructure was characterized by a scanning electron microscope (SEM) with energy disperse spectroscopy (EDS), X-ray diffraction (XRD), and X-ray photoelectron spectroscopy (XPS). The compressive mechanical properties and electrochemical corrosion behavior of the unreduced and reduced samples were discussed. Samples reduced in hydrogen at 1100 °C and 1250 °C show better compressive properties while still retaining good corrosion resistance. Reduction of oxide facilitates sintering thus improves the compressive properties. Increasing the content of solute nitrogen and reducing the precipitation of nitride can effectively improve the corrosion resistance of porous HNASS.

## 1. Introduction

High-nitrogen austenitic stainless steel has found wide applications in the fields of energy, transportation, petroleum, chemical, and medical industries, owing to its excellent combination of mechanical properties and corrosion resistance [1,2,3]. Nitrogen can enlarge the austenitic field and enhance the phase stability of austenite. Nitrogen can significantly improve the strength and corrosion resistance of austenitic stainless steel while maintaining its fracture toughness. Nitrogen can be used in stainless steel as a substitute for nickel and therefore reduce cost [3,4,5,6,7]. Austenitic stainless steel prepared via powder metallurgy route using pore forming agent has great application potential because of the advantages such as controllable porosity and pore size, and the ability to adjust the elastic modulus of the stainless steel [8].

Although a few reports have been published on sintering and nitriding of powder metallurgy stainless steel, the mechanism of oxide reduction on sintering nitriding is unknown. Zumsande et al. investigated gas-solid nitridation of austenitic steel and cold-work tool steel and detected the first gas-solid reaction at 700 °C [9]. With the powder metallurgy method, nitrogen is widely used to provide a protective atmosphere during sintering. The gas-solid reaction between nitrogen and stainless steel increases the nitrogen content in steel and this promotes the transformation of steel from ferritic to austenite phase [10].

Oxides on the powder surface prevent both the densification processes of sintering and nitriding. For instance, Borgström et al. explored the atmosphere impact on the reduction of oxides in aerosolized high-speed steel powder and compacted billet. They found that the removal of surface oxides is the premise of nitriding. Nitrogen absorption conforms to Sieverts’ law provided the oxides are removed, and the nitriding effect depends on the quantity of surface oxides [11]. Therefore, the removal of surface oxides is a key step that affects nitriding and sintering.

Badin et al. explored an approach of reducing oxides to form stainless steel with a micron porous surface by heating to 1100 °C in a hydrogen atmosphere. However, the mechanism via which the reducing atmosphere, temperature, and pressure impact the reduction and pore formation is not clear [12]. Turkdogan et al. reduced Fe_2_O_3_ and FeO under a mixed atmosphere of hydrogen and water vapor and assessed the formation of pores [13]. Elsewhere, Chu et al. established kinetic and thermodynamic models for hydrogen reduction of Cr_2_O_3_ [14,15]. Zumsande et al. found the oxides of 19Mn-18Cr steel decomposed at less than 700 °C, and the reduction of chromium oxides by carbon in the alloy began at 700 °C, accompanied by high evaporation loss of manganese [15]. Garcia-cabezon et al. [16] and Hu et al. [17] successfully fabricated porous nickel-free austenitic stainless steel under a mixed gas of nitrogen and hydrogen with the ratio of 95:5 using powder metallurgy. However, these studies did not report on the effect of hydrogen reduction on nitriding and sintering.

Although the advantages of fabricating porous high nitrogen austenitic stainless steel by powder metallurgy method are obvious, the gas-solid nitriding is a complex and slow process. Unless a special powder metallurgy process is developed, gas-solid sintering and nitriding are only suitable for sintered parts with a small thickness or porous material. In this paper, porous high-nitrogen nickel-free austenitic stainless steel was fabricated via the powder metallurgy method. Reduction and nitriding were performed under different conditions, followed by the exploration of the phase, microstructure, oxide morphology, and distribution of nitrogen elements. The influence of hydrogen reduction on compressive properties and corrosion resistance was analyzed.

## 2. Materials and Methods

Near-spherical N_2_ gas atomized duplex stainless steel powders with a particle size of 100–250 μm used are showed in Figure 1., and its chemical composition is shown in Table 1. The as-received powders were provided by Zhonglian advanced steel material technology Co., Ltd., Beijing, China.

About 30 vol.% NH_4_HCO_3_ was used as the pore-forming agent and 0.5 wt.% polyvinyl alcohol (PVA) was used as the binder [18]. The as-received stainless steel powder, pore-forming agents, and binder were mixed in a V-type mixing machine for 24 h. The evenly mixed powder was compacted under a pressure of 800 MPa and held for 1 min by using a hydraulic press machine. Vaporization of pore-forming agents and elimination of binder was carried out at 200 °C for 1 h and 400 °C for 1 h in an OTF-1200 tube furnace, then holding the samples at 900 °C for 1 h to presinter. The heating rate for all the heating processes under a flowing N_2_ atmosphere was 10 °C/min. The as-presintered samples were 15 mm in diameter and 10 mm in height. Details of the experiment and the corresponding sample codes are shown in Table 2.

A group of samples were directly sintered in a flowing N_2_ atmosphere under 1200, 1250, and 1300 °C for 2 h in a GSL-1700 tube furnace, respectively, and then the furnace was cooled. In the other group, samples were heated to 700 °C, 1100, 1200, and 1250 °C in a box furnace (KSL-1700X-H2), respectively, from room temperature under pure hydrogen atmosphere and held for 15 min. After this, the hydrogen atmosphere was replaced by N_2_ and heated to 1250 °C and held for 2 h. Then, the furnace cooled to room temperature. Both groups were heated at a heating rate of 10 °C/min.

A scanning electron microscope (SEM, NOVA NANOSEM430) was used to examine the microstructure of the powder and sintered samples. Microstructural compositions of samples were analyzed with EDS (Oxford company, Oxford, UK). Phase constituents were examined by X-ray diffractometer (XRD, Rigaku SmartLab SE, Japan) with scanning angle range of 30–100° and scanning rate of 0.04°/s. Cu Kα radiation was used. The overall N and O content in as-fabricated porous samples were measured by oxygen/nitrogen/hydrogen analyzer (ONH836, LECO, St. Joseph, MI, USA). X-ray photoelectron spectroscopy (XPS) was used to identify the elements and their chemical states on the surfaces and in the core of the samples, using a Thermo K-Alpha+ instrument equipped with an Al Kα X-ray source, operating under a vacuum of ~2 × 10^−7^ Pa and a pass energy of 30 eV to enable high resolution of the spectra. The analyzed areas were 400 μm × 300 μm. C1s peak at 284.8 eV was used to correct peak positions on the XPS spectrum. The detection depth of XPS is typically about 5–10 nm. XPSPEAK41 software was applied for XPS spectra analysis whereas the Shirley method was employed for background subtraction. The binding energy data of nitrogen element was obtained from NIST XPS Database and published papers [15,19,20,21,22]. Cross-sectional polished samples were prepared to detect core zone by XRD, SEM, and XPS. Compressing tests were performed on a universal testing machine (E45.105B, MTS systms Corporation, Eden Prairie, MN, USA) at a strain rate of 1 × 10^−3^ s^−1^, the dimension of the samples was 2 mm (diameter) × 4 mm. A compression stress–strain curve was constructed after recording force and deformation (changes in length) electronically. The experiment was repeated five times in each group. Polarization tests were carried out in 0.9 wt.% NaCl solution using an electrochemical workstation (IviumStat, Ivium Technologies BV, Eindhoven, The Netherlands) at room temperature (26 °C). The sample surface was coated with epoxy resin to expose the measurement area of 1 cm^2^. The open circuit potential was measured for 3600 s. Potentiodynamic polarization curves were measured in the potential range from −0.8 mV to 0.8 mV vs. open circuit potential (E_OCP_) at a scan rate of 1 mV/s. A reference and counter electrode were saturated Ag/AgCl and Pt electrodes, respectively [21,23,24,25].

## 3. Results and Discussion

### 3.1. Sintering in Pure Nitrogen

XRD patterns of the as-received stainless steel powder and the prepared porous high-nitrogen stainless steel are shown in Figure 2. The XRD pattern of as-received powder is presented in Figure 2a, which confirms that the powder consisted of ferrite and a slight amount of austenite.

All samples prepared without hydrogen reduction were composed of austenite, ferrite, and chromium manganese oxide. Samples sintered at 1200 °C contain mainly ferrite and with small amount of austenite and chromium manganese oxide. However, with the increase of sintering temperature, the matrix changed from a dual-phase of ferrite and austenite to mostly austenite, and the amount of oxide decreased gradually.

Figure 3 displays SEM micrographs of the core zone of N1200, N1250, and N1300 specimens. The sintering necks of the specimens are presented in Figure 3a,c,e. Large pores were formed after the volatilization of the pore-forming agents. Small pores were formed by the interstitial space between the particles. Large round holes in the grain were defects of the powder grain, but not caused during sintering. There were spot, short bar, and lamellar precipitates in the N1200 sample (Figure 3a,b). Based on the EDS results, the atomic ratio of N and Cr in these precipitates was about 1:2. It was presumed that the precipitates are Cr_2_N.

Based on the XRD results from the N1200 sample (Figure 2b), its main composition was ferrite with a little amount of austenite, and the nitrogen content was higher after nitriding compared to the as-received powder. A small amount of ferrite transformed to austenite. Nitride precipitation occurred because of excess nitrogen beyond solid solubility, but the nitride concentration was low and thus did not show a peak in the XRD pattern (Figure 2b). The boundaries of particles were noticeable (Figure 3a), indicating that the incomplete sintering is not related to the good metallurgical bonding between particles. According to the XRD pattern in Figure 2, the N1250 sample was mainly composed of austenite with a small amount of ferrite. The SEM micrographs of N1250 sample (Figure 3c,d) demonstrate that the metallurgical bonding between particles is achieved. Lamellar precipitation was the only precipitated nitride detectable at high magnification. Compared to the N1200 sample, the N1300 sample sintered better, and most of the particle boundaries disappeared. Moreover, the precipitates are mainly lamellar. The nitrogen contents of N1250 and N1300 were much higher than that of the as-received powder. The EDS results are shown in Table 3. As the sintering temperature rises, the samples tend to absorb more N under the N_2_ atmosphere, which leads to the increased N content and results in complicated nitrides. The matrix was austenite for the most part, which was conductive to the absorption of nitrogen at the sintering temperature of 1250 °C. Nitrogen content of the sample decreased slightly when the temperature was raised from 1250 °C to 1300 °C.

### 3.2. Effect of Reduction by H_2_

The XRD pattern of hydrogen reduced samples of porous duplex stainless steel is shown in Figure 4. All four groups of samples showed the dual-phase of austenite and a small amount of ferrite, accompanied by little nitride precipitation.

The SEM micrograph of sample H700 is shown in Figure 5a,b. Large pores were formed by the elimination of pore-forming agents. Small pores were formed by the reduction of metal oxides. The iron oxide could be reduced in hydrogen at 700 °C [26,27]. Space was formed after reduction, which developed into a small hole (Figure 5a). The precipitates were mainly lamellar Cr_2_N which is diffused.

The overall nitrogen and oxygen content of the samples are displayed in Table 4. The oxygen content of samples H1100 was 0.14 wt.%, the lowest compared to other reduced samples. Moreover, the nitrogen content reached 2.25 wt.%. SEM micrographs shown in Figure 5c,d demonstrate the precipitates are finely lamellar, parallel, and evenly distributed in the crystal particle interior. The particle surface of sample H1200 is covered by a layer of oxide shown in Figure 5e, which we identified, using EDS (shown in Table 5), to be MnO. The formation of Mn oxide in the surface of particle hindered the sintering process and reduced the Mn content of matrix.

The sintering of sample H1250 was favorable and its overall oxygen content was less than that of sample H1200. The precipitates were fine and evenly distributed (Figure 5h), the interval of lamellar space was less than 1 mm. Three types of pores exist in the samples reduced by hydrogen. Type 1 pores left by pore-forming agents with the size of 50–300 μm; type 2 pores produced by gap between particles with the size of 10–50 μm; type 3 pores left by oxide reduction with the size less than 1 μm, are mostly distributed at the junctions of the particles. Type 2 and type 3 pores with large sizes are conductive to complete the reduction. The pores can provide access for reducing gas H_2_ which can come into contact with the stainless steel particles and promote the diffusion out of the locally generated gaseous products. Meanwhile, the existence of pores also provide access for N_2_ to accelerate the gas–solid nitridation.

Figure 6a–d compare the existing states of N elements in non-reduced sample N1250 and reduced samples. The N 1s peak is composed of four peaks, CrN (396.7 eV), Cr_2_N (397.4 eV), solute N (398.1 eV) [19,20,21,22], and chromium nitrates CrN_x_O_y_ (399.4 eV) [28,29]. The quantitative results of the N content tested by XPS on various samples are shown in Table 6. The results of total N content in Table 6 were obtained by using an oxygen/nitrogen/hydrogen analyzer. There are two states of nitrogen elements for stainless steel samples sintered with high-temperature gas-solid nitriding: solid solution state and combined state (nitrides and chromium nitrates) [22], which are mainly precipitated in the process of slow cooling in the furnace. The solute N content of N1250 is highest among the tested samples.

### 3.3. Effect of Reduction on Compressive Properties and Corrosion Resistance

The compressive engineering stress–strain curves of the as-fabricated HNASSs are shown in Figure 7. Table 7 lists the compressive strength, compressive stain, and porosity of the samples. The measurements of porosity were carried by Archimedes method. As shown in Table 7, with the increase of sintering temperature, the porosity decreases slightly, compressive strength and strain increase gradually because of better sintering effect on the one hand and lower porosity on the other hand. Reduced samples H1100 and H1250 exhibited higher compressive strength and strain compared to N1250. The reducing temperature of H700 was too low to reduce Mn oxide and Cr oxide that hindered sintering and nitriding process (content of solute nitrogen is low). An oxide layer on the particles surface was formed in the sample H1200, which greatly damages the compression performance.

Figure 8 shows potentiodynamic polarization curves measured in 0.9 wt.% NaCl. All the tested specimens show a similar passivation behavior. The corresponding fitting parameters are shown in Table 8. Samples sintered without hydrogen reduction showed lower Ecorr compared to the reduced samples. The corrosion current density of N1250 is lowest (8.767 × 10^−5^ A/cm^2^) because of the highest content of solute N. The corrosion potential of H1100 is the highest (−0.6489 eV) among other samples indicating lower corrosion susceptibility due to the low content of nitride (CrN and Cr_2_N). Low corrosion potential of the as-fabricated porous samples was mainly from the porous structure and nitride content of the microstructure. On the one hand, the precipitation of chromium nitride reduces Cr content in the matrix, thus decreasing the electrode potential of the matrix. On the other hand, many precipitates are distributed in the matrix which leads to intergranular corrosion [21,22].

SEM morphology of as-fabricated porous samples H700 and H1100 after polarization tests was shown in Figure 9. Cracks are likely to form in sample H700 after polarization tests (shown in Figure 9a) due to the oxide inclusions causing weakness at the particle boundary. Overall corrosion resistance of sample H1100 is better than H700, and no obvious crack emerged in the particle joints of sample H1100.

Therefore, to obtain the porous high-nitrogen austenitic stainless steel with excellent performance, the nitride content can be controlled by optimizing the nitriding process and regulating the content of nitrogen element.

## 4. Conclusions

The present study fabricated samples of porous high-nitrogen nickel-free austenitic stainless steel through powder metallurgy with the reduction in hydrogen. In addition, the effect of hydrogen reduction on compressive and corrosion properties has been investigated. 

Part of the matrix changed from ferrite to austenite when sintered in pure N_2_. The matrix is austenite for the most part and is conductive to the absorption of nitrogen at 1250 °C and 1300 °C sintering temperature. In a pure hydrogen atmosphere, as the reduction temperature increases from 700 °C to 1100 °C, the oxygen content of the sample decreases while the nitrogen content increases. The microstructure mainly is composed of austenite and a small amount of ferrite. Sample H1100 shows highest compressive strength (673.6 MPa) compared to other samples and still retains good corrosion resistance.

The hydrogen reduction reduces the oxygen content of the samples and improves the compressive properties of the material. The corrosion resistance of the material can be improved by increasing the content of solute nitrogen and reducing the formation of nitride.

## Figures and Tables

**Figure 1 materials-15-05585-f001:**
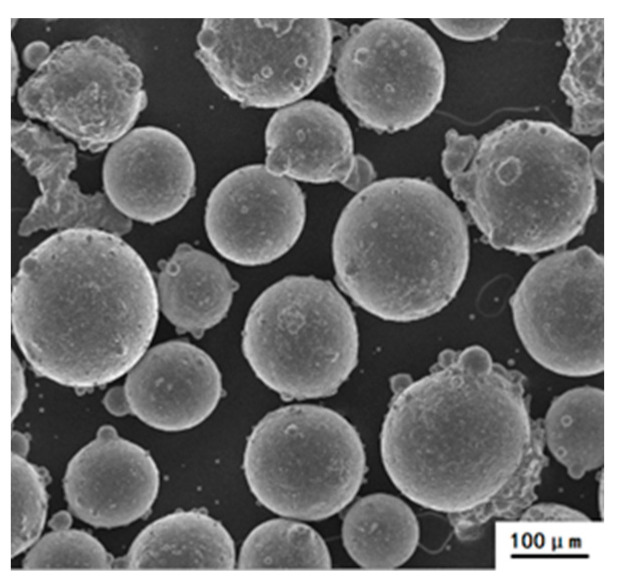
SEM morphology of the as-received stainless steel powders.

**Figure 2 materials-15-05585-f002:**
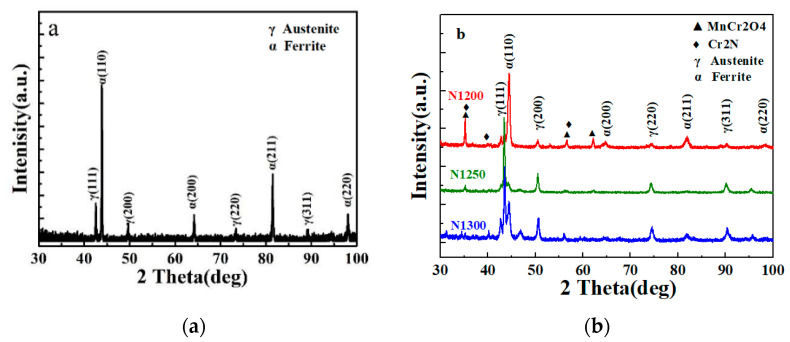
XRD patterns of (**a**) as-received powders; (**b**) porous HNASS samples of N1200, N1250 and N1300.

**Figure 3 materials-15-05585-f003:**
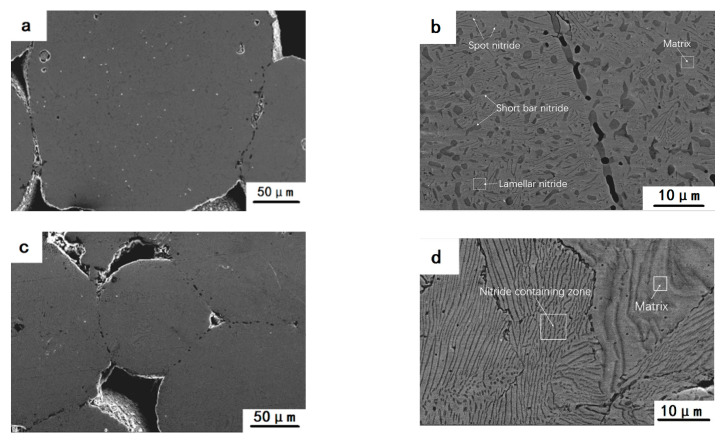
SEM micrographs of as-fabricated samples: (**a**,**b**) N1200; (**c**,**d**) N1250; (**e**,**f**) N1300.

**Figure 4 materials-15-05585-f004:**
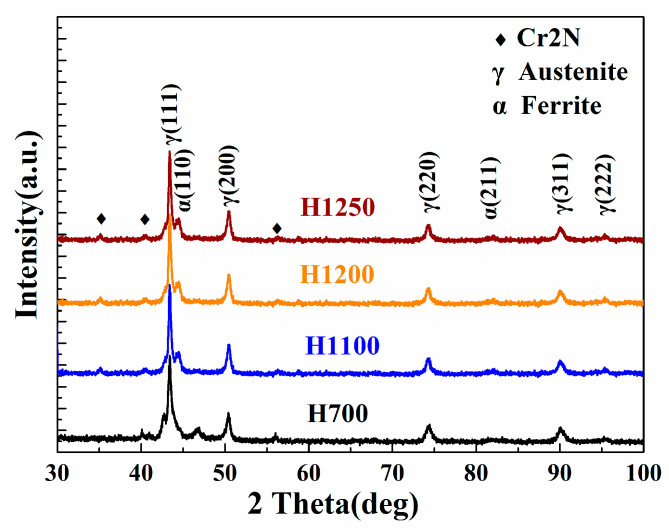
XRD patterns of porous HNASSs reduced by H_2_.

**Figure 5 materials-15-05585-f005:**
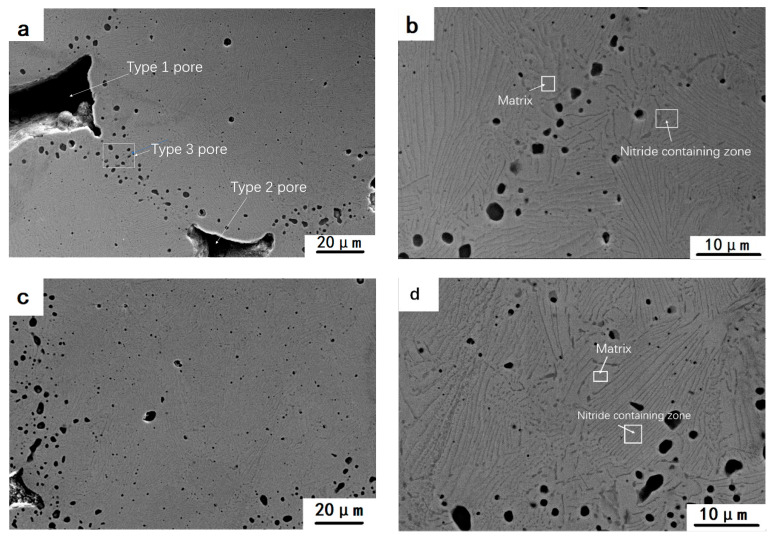
SEM morphology of porous samples reduced by H_2_: (**a**,**b**): H700; (**c**,**d**): H1100; (**e**,**f**): H1200; (**g**,**h**):H1250.

**Figure 6 materials-15-05585-f006:**
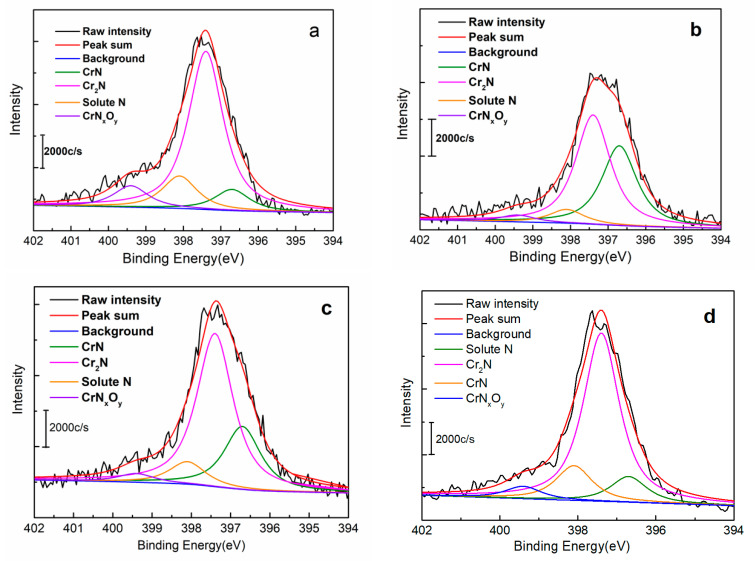
XPS peak convolution of CrN, Cr_2_N and solute N in the as-fabricated porous samples: (**a**) N1250; (**b**) H700; (**c**) H1100; (**d**) H1250.

**Figure 7 materials-15-05585-f007:**
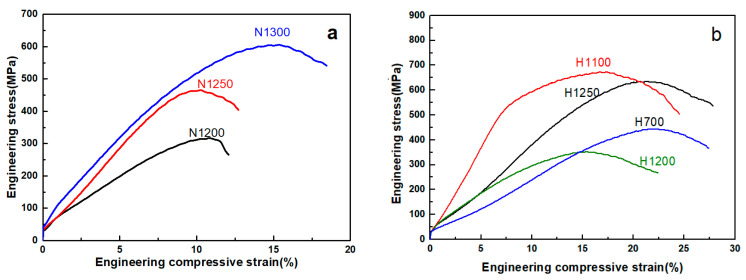
Engineering compressive stress–strain curve of the as-fabricated porous HNASS samples. (**a**) unreduced samples N1200, N1250 and N1300, (**b**) reduced samples H700, H1100, H1200 and H1250.

**Figure 8 materials-15-05585-f008:**
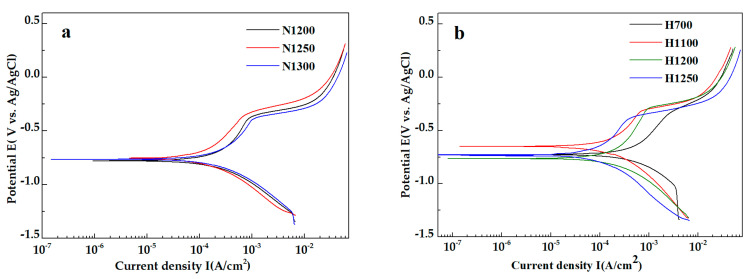
Potentiodynamic polarization curves of the as-fabricated porous samples in 0.9 wt.% NaCl solution. (**a**) unreduced samples N1200, N1250 and N1300, (**b**) reduced samples H700, H1100, H1200 and H1250.

**Figure 9 materials-15-05585-f009:**
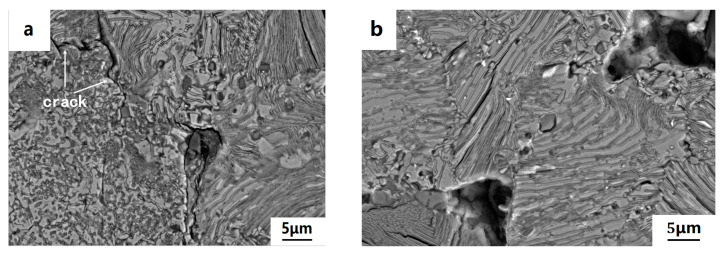
SEM morphology of porous samples reduced by H_2_ after polarization tests: (**a**) H700; (**b**) H1100.

**Table 1 materials-15-05585-t001:** Chemical composition of as-received stainless steel powders.

Element	C	Cr	Mn	N	Ni	Si	Fe
Content (wt.%)	0.08	20.34	15.45	0.15	0.10	0.55	Bal.

**Table 2 materials-15-05585-t002:** Sample codes and corresponding processing parameters (the sintering time is 2 h).

Sample Codes	Reduction Temperature in H_2_ (°C)	Reduction Time (min)	Sintering Temperature in N_2_ (°C)
N1200	-	-	1200
N1250	-	-	1250
N1300	-	-	1300
H700	700	15	1250
H1100	1100	15	1250
H1200	1200	15	1250
H1250	1250	15	1250

**Table 3 materials-15-05585-t003:** EDS results of porous samples N1200, N1250, and N1300, wt.%.

Analyzed Zone	N	O	Si	Cr	Mn	Fe
N1200 matrix	0.17 ± 0.12	0.12 ± 0.10	0.63 ± 0.08	14.63 ± 1.41	13.99 ± 0.39	Bal.
N1200 nitride containing zone	4.44 ± 0.40	0.56 ± 0.18	0.44 ± 0.03	26.53 ± 1.24	12.92 ± 0.31	Bal.
N1200 nitride	9.74 ± 0.35	0.65 ± 0.15	1.25 ± 0.13	67.74 ± 2.12	10.12 ± 0.33	Bal.
N1250 matrix	0.96 ± 0.19	0.58 ± 0.10	0.63 ± 0.03	19.17 ± 1.21	13.82 ± 0.42	Bal.
N1250 nitride containing zone	3.08 ± 0.09	0.73 ± 0.14	0.59 ± 0.03	22.55 ± 0.26	13.05 ± 0.40	Bal.
N1300 matrix	0.82 ± 0.06	0.45 ± 0.09	0.49 ± 0.08	16.91 ± 1.39	13.32 ± 0.32	Bal.
N1300 nitride containing zone	3.29 ± 0.11	0.59 ± 0.10	0.41 ± 0.09	22.87 ± 1.20	12.27 ± 0.13	Bal.

**Table 4 materials-15-05585-t004:** The overall nitrogen and oxygen contents of porous samples reduced by H_2_.

Sample Code	Measured Value	
	N (wt.%)	O (wt.%)
H700	2.11	0.38
H1100	2.25	0.14
H1200	2.22	0.54
H1250	2.19	0.40

**Table 5 materials-15-05585-t005:** The EDS results of as-fabricated porous samples reduced by H_2_, wt.%.

Analyzed Zone	N	O	Cr	Mn	Fe
H700 nitride containing zone	3.23 ± 0.24	-	22.10 ± 0.21	13.68 ± 0.08	Bal.
H700 matrix	0.57 ± 0.02	-	19.11 ± 0.15	14.32 ± 0.12	Bal.
H1100 nitride containing zone	3.63 ± 0.13	-	22.37 ± 0.22	13.49 ± 0.25	Bal.
H1100 matrix	0.69 ± 0.09	-	19.94 ± 1.32	14.49 ± 0.40	Bal.
H1200 matrix	0.91 ± 0.06	-	18.98 ± 0.32	13.64 ± 0.21	Bal.
H1200 nitride containing zone	2.72 ± 0.12	0.66 ± 0.08	22.68 ± 0.08	12.68 ± 0.41	Bal.
H1200 oxide containing zone	0.48 ± 0.12	24.69 ± 2.69	1.76 ± 1.13	71.31 ± 5.27	Bal.
H1250 matrix	0.86 ± 0.15	-	19.38 ± 1.52	11.82 ± 0.51	Bal.
H1250 nitride containing zone	3.54 ± 0.22	-	23.11 ± 0.50	12.65 ± 0.51	Bal.

**Table 6 materials-15-05585-t006:** Nitrogen content in different states for the porous HNASSs (wt.%).

Sample	CrN	Cr_2_N	Solute N	CrN_x_O_y_	Total N
N1250	0.25	1.83	0.38	0.24	2.7
H700	0.80	1.10	0.14	0.07	2.11
H1100	0.57	1.39	0.21	0.08	2.25
H1250	0.21	1.53	0.32	0.13	2.19

**Table 7 materials-15-05585-t007:** Compressive strength and elastic modulus of porous HNASS samples.

Sample Codes	Compressive Strength (MPa)	Compressive Strain (%)	Porosity (%)
N1200	311.7 ± 16.1	11.3 ± 0.8	28.6 ± 0.6
N1250	465.4 ± 10.3	10.3 ± 0.7	27.9 ± 0.4
N1300	605.8 ± 30.0	15.4 ± 1.2	25.6 ± 0.8
H700	443.6 ± 17.2	21.6 ± 0.5	27.6 ± 0.8
H1100	673.6 ± 15.6	16.7 ± 0.9	27.1 ± 0.5
H1200	352.4 ± 11.7	15.4 ± 0.4	28.2 ± 0.3
H1250	635.9 ± 18.5	21.3 ± 1.1	27.5 ± 0.6

**Table 8 materials-15-05585-t008:** Corrosion parameters of the as-fabricated porous HNASS samples obtained from the potentiodynamic polarization curves.

Sample	OCP (V vs. Ag/AgCl)	Ecorr (V vs. Ag/AgCl)	Icorr. (A/cm^2^)	R (Ω)	Ba V/dec	Bc V/dec	Corrosion Rate (mm/y)
N1200	−0.7794	−0.8107	2.439 × 10^−4^	1501	0.803	0.286	0.1478
N1250	−0.7366	−0.7374	8.767 × 10^−5^	3639	0.494	0.292	0.05312
N1300	−0.7704	−0.7721	2.744 × 10^−4^	1266	0.547	0.315	0.1662
H700	−0.7256	−0.7191	4.745 × 10^−4^	451.2	0.531	0.339	0.2875
H1100	−0.6489	−0.6412	1.848 × 10^−4^	2183	0.580	0.387	0.112
H1200	−0.7635	−0.7568	2.252 × 10^−4^	1818	0.752	0.343	0.1364
H1250	−0.7189	−0.7098	1.077 × 10^−4^	3809	0.622	0.381	0.06524

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
