# Peer review of "Influence of Hydrogen Reduction on the Properties of Porous High-Nitrogen Austenitic Stainless Steel"

_materials, 2022, doi:10.3390/ma15165585_

Round 1
Reviewer 1 Report
This paper presented the impact of hydrogen reduction on sintering and nitriding of porous high-nitrogen austenitic stainless steel processed via powder metallurgy. However, the following minor issues should be addressed before publication.
Introduction:
Lines 29 and 30, how do porosity and pore size controlled through powder metallurgy? Can the authors cite some papers here.
Line 46/47 what do authors mean by micron porous….. ? micro porous might be better.
Line 49: Correct formular is Fe2O3 not Fe2O3
Line 52: Cr2O3 should be written Cr2O3
Materials:
Figure 1 is too blurry. Can you please a clear image?
Line: 80 …..by using an hydraulic press…, “an” should be relace with “a”.
Line 82: …….furnace prior to holding the compacted powder at 900 ËšC. Please re-write this sentence
Line 87: ……….. A group of samples were directly nitridation sintered in a nitrogen…. What do authors mean by nitridation sintered??
Results:
Add more detail to all Figure captions. It should describe the fully describe the figures and stand alone.
Line 139: There were spot, short bar, and lamellar precipitates in the N1200 sample (Figure 3a) and b)). Can you please indicate spot, short bar and Lppt in the SEM images?
Be consistent in citing Figures. Some places you use short form like Fig. 2b and full in some places.
Space 172
Lines 172/173: Space was formed after reduction, which developed into a small hole (Figure 5 a)). Kindly indicates all features in the SEM images such as small hole you stated.
Be consistent in writing terms. For instance, you wrote Type1 in line 186 and Type 2 in line 183.
Add 2021 and 2022 references.
There are some grammatical errors. The paper need to be fully proofread.
Author Response
Point by point reply
We are grateful for the reviewers’ assessment of our work and deeply apprecciate their thoughtful comments and constructive suggestions, which we have fully addresed in this revisison and believe that this has strengthened the manuscript significantly. Our changes in the manuscript in red.
Lines 29 and 30, how do porosity and pore size controlled through powder metallurgy? Can the authors cite some papers here.
Reply: Different size and amount of pore forming agents were used to control pore size and porosity through powder metallurgy. Relevant papers have been cited.
Line 46/47 what do authors mean by micron porous….. ? micro porous might be better.
Line 49: Correct formular is Fe2O3 not Fe2O3
Line 52: Cr2O3 should be written Cr2O3
Reply: Thanks to the reviewer 1# for valuable suggestions, the above editorial-type deficiencies have been modified.
Materials:
Figure 1 is too blurry. Can you please a clear image?
Reply: It has now been modified in the revised manuscript.
Line: 80 …..by using an hydraulic press…, “an” should be relace with “a”.
Reply: It has now been modified in the revised manuscript.
Line 82: …….furnace prior to holding the compacted powder at 900 ËšC. Please re-write this sentence
Reply: It has now been re-written in the revised manuscript.
Line 87: ……….. A group of samples were directly nitridation sintered in a nitrogen…. What do authors mean by nitridation sintered??
Reply: A group of samples were sintered in a flowing N2 for sintering and nitridation.
Results:
Add more detail to all Figure captions. It should describe the fully describe the figures and stand alone.
Reply: We have added more detail to describe the Figures.
Line 139: There were spot, short bar, and lamellar precipitates in the N1200 sample (Figure 3a) and b)). Can you please indicate spot, short bar and Lppt in the SEM images?
Reply: We have labeled the different precipitates in the SEM images.
Be consistent in citing Figures. Some places you use short form like Fig. 2b and full in some places.
Reply: It has now been modified in the revised manuscript.
Lines 172/173: Space was formed after reduction, which developed into a small hole (Figure 5 a)). Kindly indicates all features in the SEM images such as small hole you stated.
Reply: We have labeled the small hole (Type 3 pore) in the SEM images.
Be consistent in writing terms. For instance, you wrote Type1 in line 186 and Type 2 in line 183.
Reply: We have labeled the different pore in the SEM images.
Add 2021 and 2022 references.
Reply: 2021 and 2022 references are added in the revised manuscript.
There are some grammatical errors. The paper need to be fully proofread.
Reply: The paper have been fully proofread.

Reviewer 2 Report
The manuscript is interesting; however, some points need to be corrected or extended. More details can be found in the attached file.

Author Response
Point by point reply
We are grateful for the reviewers’ assessment of our work and deeply apprecciate their thoughtful comments and constructive suggestions, which we have fully addresed in this revisison and believe that this has strengthened the manuscript significantly. Our changes in the manuscript in red.
Line 49 Please use subscript for digit here and further in the work.
Reply: It has now been modified in the revised manuscript.
Line 64 Did you use pure nitrogen or a mixture with hydrogen? Hydrogen is well known as an oxide-reducing gas in the case of metals. It is generally used because it is ambient for the metal.
Reply: We used pure nitrogen and hydrogen. The samples were reduced in pure H2, then sintered in pure N2.
Line66 Hydrogen or nitrogen?
Reply: Hydrogen reduction.
Line75 Please add the cross-section of the etched powder to analyse the structure.
The particles were produced by gas atomized with homogenous composition and structure, and it is difficult to add the cross-section of the etched powder.
Line81 The elimination of binder was carried out in air or nitrogen atmosphere?
Reply: The elimination of binder was carried out in nitrogen atmosphere. Line 85 has described。
Line92 What was the cooling time?
Reply: The samples cooled with furnace cooled. Line 89 and line 93 have described.
Line111 Please provide a standard or give more details of compressing tests.
Reply: More details were added in the revised manuscript.
Line 113 The NaCl concentration is very small, usually it is 3,5-5%. Why do you underrate the value? It is obvious that the samples will stay in it unchanged longer... ?
Reply: High nitrogen was used in stainless steel as a substitute for nickel to reduce the harm to human body in medical applications. Thus 0.9 wt.% NaCl solution was used to simulate the environment of human body fluids.
Line120 It should be results and DISCUSSION. In the text below, there is only presentation of results, without any discussion. I strongly recommend that you provide a discussion.
Reply: More discussion and data were added in the revised manuscript.
Table 4 Why is there a difference in oxygen content in sample H1200? In my opinion, you should repeat sample preparation.
Reply: We had repeated sample preparation. We are also curious about the reasons for the formation of manganese oxide, but we have not found literature of analogue for the time being. It may be related to the effect of reducing atmosphere on Mn diffusion at this temperature, which requires further research.
Table 5 What is the difference between tab 4 and tab 5 ? The method, place? It is not clear.
Reply: Table 4 shows the EDS results of local position. Table 5 shows the overall N and O contents of the as-fabricated porous samples obtained from Oxygen/ Nitrogen/ Hydrogen Analyzer (ONH836, LECO, USA).
Table 7 Therefore, I suggest repeating preparation of sample H1200.
Reply: n oxide layer on the particles surface was formed in the sample H1200, which greatly damages the compression strength.
Line 233 In my opinion you should add a cross section of the best and the worst samples after polarization tests to compare the structure.
Reply: The SEM micrographs of cross section of the best and worst samples after polarization tests were added in the revised manuscript.
Round 2
Reviewer 2 Report
After revision the manuscript can be published in the journal.